# IODA: Instance-Guided One-shot Domain Adaptation for Super-Resolution

**Zai-Zuo Tang, Yu-Bin Yang**[*]
State Key Laboratory for Novel Software Technology
Nanjing University
Nanjing, China
`tangzz@smail.nju.edu.cn, yangyubin@nju.edu.cn`

## Abstract

The domain adaptation method effectively mitigates the negative impact of domain gaps on the performance of super-resolution (SR) networks through the guidance of numerous target domain low-resolution (LR) images. However, in real-world scenarios, the availability of target domain LR images is often limited, sometimes even to just one, which inevitably impairs the domain adaptation performance of SR networks. We propose Instance-guided One-shot Domain Adaptation for Super-Resolution (IODA) to enable efficient domain adaptation with only a single unlabeled target domain LR image. To address the limited diversity of the target domain distribution caused by a single target domain LR image, we propose an instance-guided target domain distribution expansion strategy. This strategy effectively expands the diversity of the target domain distribution by generating instance-specific features focused on different instances within the image. For SR tasks emphasizing texture details, we propose an image-guided domain adaptation method. Compared to existing methods that use text representation for domain difference, this method utilizes pixel-level representation with higher granularity, enabling efficient domain adaptation guidance for SR networks. Finally, we validate the effectiveness of IODA on multiple datasets and various network architectures, achieving satisfactory one-shot domain adaptation for SR networks. Our code is available at https://github.com/ZaizuoTang/IODA.

## 1 Introduction

Super-resolution (SR) has significant practical value in various fields, including remote sensing imagery [3, 4, 5], the restoration of old films [6, 7], and even as a preprocessing strategy to enhance downstream tasks such as object detection [8, 9, 10] and image segmentation [11, 12, 13]. However, SR also faces domain gap issues, where the performance of networks significantly declines when the distribution of test low-resolution (LR) images differs from that of the training LR images [14]. Therefore, domain adaptation methods for SR [14, 15] have emerged. These methods represent training LR images as the source domain and test LR images

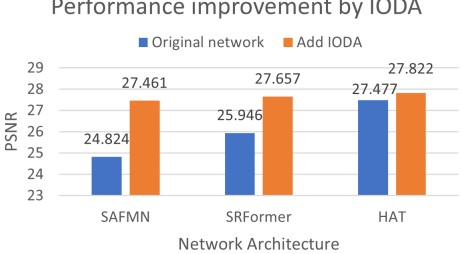

Figure 1: Performance improvement by IODA. The performance of the real-world pre-trained weights (from DF2K dataset [1]) provided in the original paper serves as the baseline on the real-world dataset RealSR_Canon [2].

---

[*]corresponding author

38th Conference on Neural Information Processing Systems (NeurIPS 2024).

as the target domain, then use generative adversarial networks to align the two domains during adaptation training, mitigating the negative impact of domain gaps on network performance. (Refer to Appendix A.2 for related work on SR network.)

However, the methods mentioned above require a large number of target domain LR images for training the generative adversarial network alignment. In some special scenarios, such as photos taken by users or uploaded to the internet, users often provide only a small amount or even a single LR image for SR, rendering the aforementioned methods inapplicable.

In other tasks such as image generation [16, 17, 18, 19], semantic segmentation [20], and object detection [21], one/zero-shot domain adaptation methods have emerged (collectively referred to as text-guided domain adaptation, Appendix B). These methods use text descriptions to guide domain adaptation, inputting the text descriptions of the source and target domains into the CLIP model [22] to compute the text domain difference, namely the text cross-domain direction vector. Subsequently, the predicted target domain images and source domain images are input into the CLIP model to generate the image cross-domain direction vector. By constraining the target domain generation model to align the image cross-domain direction vector with the text cross-domain direction vector, these methods achieve domain adaptation with one/zero-shot learning. (Refer to Appendix A.1 for related work on one/zero-shot domain adaptation.)

However, when it comes to the SR task, one/zero-shot domain adaptation methods have not yet been explored. If existing one/zero-shot domain adaptation methods are directly applied to the SR task, they will encounter the following two issues:

- Text descriptions lack fine granularity, which prevents accurate and comprehensive representation of domain differences in image texture details, thereby failing to provide effective guidance for domain adaptation in the SR task.

- The diversity of the target domain distribution is limited. Existing one/zero-shot domain adaptation methods typically sample latent vectors from random noise for subsequent image generation, enriching the target domain distribution through multiple samplings. However, for the SR task, the target domain consists of only a single LR image, resulting in a limited diversity of target domain distribution, which increases the risk of pattern collapse during training.

To address the first issue, this paper proposes an image-guided domain adaptation method that transforms the existing text-level description of domain differences between source and target domains into a pixel-level description of domain differences in LR images. This approach achieves a high level of granularity in representing image texture details, effectively guiding domain adaptation for SR networks. Regarding the second issue, we propose an instance-guided target domain distribution expansion strategy. By enabling Alpha-CLIP [23] to generate instance-specific features, the diversity of the target domain feature distribution is effectively expanded through the diversity of instances in the image.

To the best of our knowledge, this is the first work applying a one-shot domain adaptation method in the SR task. Our main contributions can be summarized as follows:

- We propose an image-guided domain adaptation method to address the issue of incomplete image texture detail representation in existing text-guided domain adaptation methods.

- We propose an instance-guided target domain distribution expansion strategy to address the issue of limited target domain distribution diversity caused by one-shot scenarios.

- We introduce a novel one-shot domain adaptation method for the SR task. This method utilizes a single LR image from the target domain, effectively alleviating the negative impact of domain gaps on the performance of SR networks.

- We propose a plug-and-play domain adaptation method for SR task, which demonstrates good robustness for different data distributions and network architectures.

## 2 Proposed method

### 2.1 Problem setting

Given a source domain dataset $D^{Source} = \{LR_i^{Source}, HR_i^{Source}\}_{i=1}^{N_s}$, a target domain dataset $D^{Target} = \{LR_i^{Target}, HR_i^{Target}\}_{i=1}^{N_t}$, a source domain SR network $M_S$ trained solely on the source domain dataset, and a target domain SR network $M_T$ initialized by $M_S$. The problem addressed in this paper is to adjust the network $M_T$ (one-shot domain adaptation) using only a single LR image $LR_{i-th}^{Target}$ from the target domain dataset $D^{Target}$ (without a corresponding HR image $HR_{i-th}^{Target}$ as a label), aiming to achieve better performance of the adjusted network $M_T$ on the target domain dataset $D^{Target}$ compared to the source domain network $M_S$. Notably, the target domain HR images $HR^{Target}$ are used only for performance testing after the domain adaptation training is complete and are not visible to the network during domain adaptation training.

### 2.2 Overview

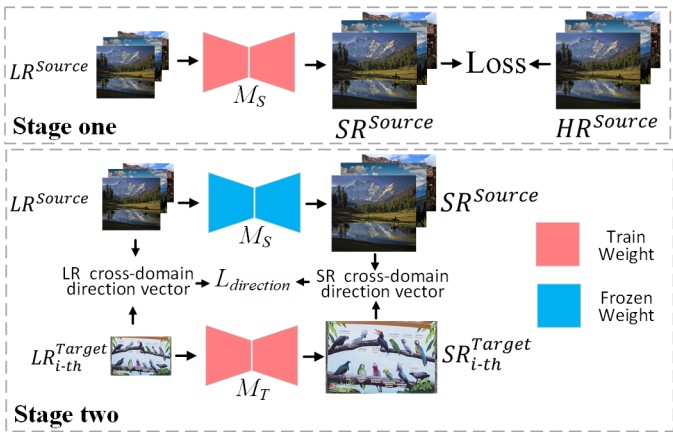

Figure 2: Overall framework

As shown in Figure 2, IODA is primarily divided into two stages. In the first stage, the SR network $M_S$ is pre-trained using the source domain dataset $D^{Source}$. In the second stage, domain adaptation is performed on the SR network. Firstly, the weight of the pre-trained SR network $M_S$ is frozen and used to initialize the target domain SR network $M_T$. Subsequently, using LR images from the source domain $LR^{Source}$ and target domain $LR_{i-th}^{Target}$, as well as SR images from the source domain $SR^{Source}$ and target domain $SR_{i-th}^{Target}$, we compute SR and LR cross-domain direction vectors in the Alpha-CLIP space. The $L_{direction}$ loss is calculated to align these two cross-domain direction vectors, enabling domain adaptation training for the target domain SR network $M_T$. The focus of this paper is on the second stage: domain adaptation of the SR network, corresponding to Sections 2.3-2.4. Subsequent sections will not delve into the details of the first stage, which aligns with the training methods employed in the original papers [24, 25, 26].

### 2.3 Image-guided domain adaptation

Existing text-guided domain adaptation methods seek the text cross-domain direction vector by comparing the domain difference between the source domain text description and the target domain text description ("cat"and"dog", as shown in Appendix B). However, text description cannot capture the domain differences in the texture details between the source and target domains, such as the orange short fur of a cat versus the black long fur of a dog, or the pink nose of a cat versus the black nose of a dog. In the SR task, texture details are extremely critical. The SR network needs to finely reconstruct texture details such as fur, eyes, and nose. Moreover, the commonly used PSNR performance metric is also calculated at the pixel level. Therefore, existing text-guided domain adaptation methods are not suitable for SR tasks (Table 2). We aim to develop a domain difference representation method with higher granularity.

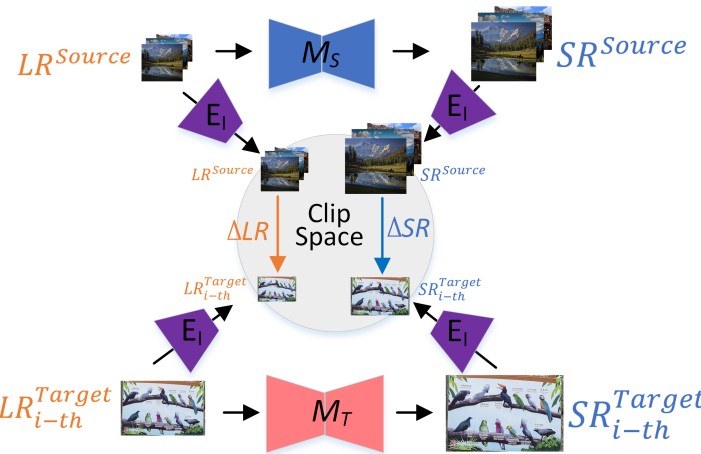

Figure 3: Image-guided domain adaptation method. This figure depicts the second stage of domain adaptation. $M_S$ and $M_T$ represent the source domain SR network and the target domain SR network, respectively.

When it comes to representing texture details, images are superior to text because they can depict texture detail features at the pixel level. Compared to the network's perception of differences between two words, the differences between images are more fine-grained and contain more detailed features. Additionally, the SR processing does not alter the image's content information. The cross-domain direction from the source domain to the target domain in LR images should also be followed in SR images (aligned with it). Specifically, in the LR image space, if transitioning from a cat image in the source domain to a dog image in the target domain, then after image SR processing, this transformation rule should also be followed: transitioning from a cat image in the source domain to a dog image in the target domain. Therefore, we propose an image-guided domain adaptation method (Figure 3), which utilizes the LR images from the source and target domains to determine the domain differences. This method has pixel-level domain difference representation, providing higher granularity and more efficient domain adaptation guidance for SR network.

As shown in Figure 3, image-guided domain adaptation directly uses the CLIP model to learn the LR image cross-domain direction vector $\Delta LR$ between the source domain LR image $LR^{Source}$ and the target domain LR image $LR_{i-th}^{Target}$. Simultaneously, it computes the SR image cross-domain direction vector $\Delta SR$ between the source domain SR image $SR^{Source}$ and the target domain SR image $SR_{i-th}^{Target}$. By imposing collinearity constraints on these two cross-domain direction vectors, it enables domain adaptation training for the target domain SR network,

$$\Delta LR = E_I(LR_{i-th}^{Target}) - E_I(LR^{Source}), \tag{1}$$

$$\Delta SR = E_I(M_T(LR_{i-th}^{Target})) - E_I(M_S(LR^{Source})), \tag{2}$$

$$L_{direction} = 1 - \frac{\Delta LR \cdot \Delta SR}{|\Delta LR| \cdot |\Delta SR|}, \tag{3}$$

$E_I$ represents the image encoder of CLIP [22].

## 2.4 Instance-guided target domain distribution expansion strategy

The one/zero-shot domain adaptation method of the generation network can construct the diversity distribution of the target domain (latent vectors) by multiple sampling from random noise (Appendix B). However, for the SR task, it is not possible to sample from random noise, as there is only one target domain LR image available. Although randomly cropping subregions from a single target domain LR image can slightly increase the diversity of the target domain distribution (as shown in Figure 4a), the diversity of the target domain distribution is still limited. These limitations increase the risk of pattern collapse during domain adaptation training, negatively impacting the efficiency of domain adaptation.

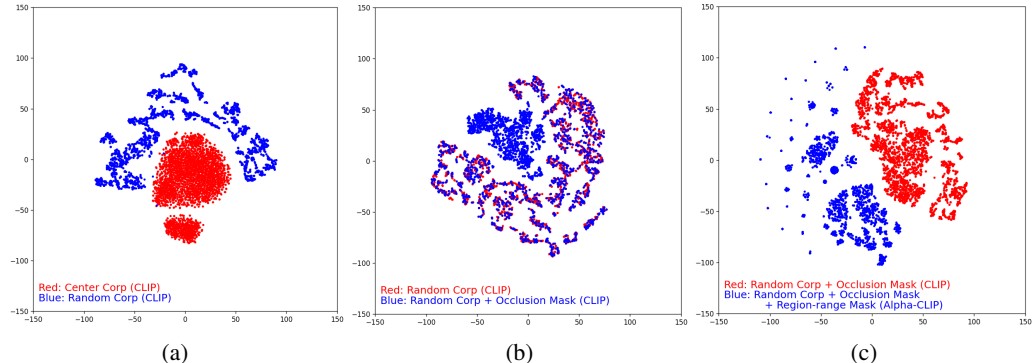

(a)                                    (b)                                    (c)

Figure 4: Feature distribution comparison. Inputting a single target domain LR image into the image encoders of the CLIP model (a, b) and the Alpha-CLIP model (c), visualizing the output features after dimension reduction using T-SNE [27] (repeated 1000 times). It's worth noting that the more dispersed the scatter plot distribution, the more diverse the target domain feature distribution.

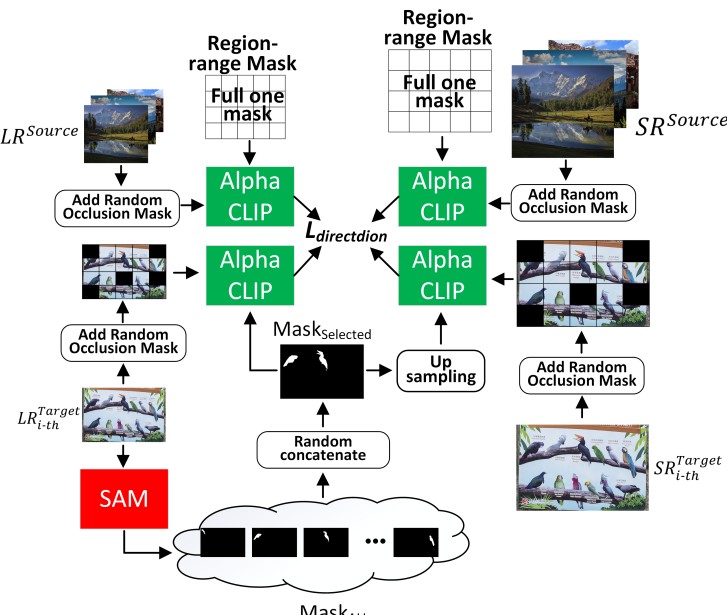

Figure 5: Instance-guided target domain distribution expansion strategy. The source domain and target domain SR images $SR^{Source}$, $SR^{Target}_{(i-th)}$ are generated from the corresponding LR images $LR^{Source}$, $LR^{Target}_{(i-th)}$ through their respective domain-specific SR networks $M_S$ and $M_T$.

Inspired by MAE [28], we intend to introduce occlusion masks to randomly mask the cropped target domain LR images and use the masked LR images for domain adaptation training of the target domain SR network. This method enhances the diversity of the target domain distribution through the randomness of the occlusion masks (as shown in Figure 4b).

Although occlusion masks increase the diversity of target domain distribution to some extent, we aim to further expand the extent of this increasement. Recently, Alpha-CLIP [23] (as shown in Appendix C) has received widespread attention for its ability to enhance CLIP [22] by incorporating specified attention regions. By providing Alpha-CLIP with a region-range mask, the network can focus more on the specified area. Motivated by this, we intend to provide Alpha-CLIP with different region-range masks each time, enabling it to generate Alpha-CLIP spatial features focused on different areas of the image, thereby expanding the diversity of the target domain distribution.

For the generation strategy of region-range masks, benefiting from the powerful performance of the SAM model [29], which has good generalization and can accurately segment each instance in arbitrary scene images. SAM is used to generate region-range masks, corresponding to each instance in the image. Specifically, we use region-range masks generated by SAM to enable Alpha-CLIP to focus on different instances when generating multiple Alpha-CLIP spatial features (instance-specific features). For example, in Figure 5, features could focus on the leaves, the first bird, the second bird, and so on. In addition, compared to naively randomly generating rectangular region masks, instance-specific masks implicitly convey semantic information, exhibiting better domain adaptation efficiency (Table 3). As shown in Figure 4c, the introduction of Alpha-CLIP equipped with region-range mask further enhances the diversity of the target domain distribution.

As shown in Figure 5, we replace the CLIP image encoder with the Alpha-CLIP image encoder. Firstly, we use the SAM model to perform instance segmentation on the target domain LR image, obtaining a collection of instance masks, denoted as $Mask_{ALL} \in \{mask_i\}_{i=1}^{N_M}$. Randomly select a certain number of masks from the mask collection to concatenate and generate region-range mask $Mask_{Select}$. The target domain LR images with added occlusion mask and region-range mask $Mask_{Select}$ are input into the image encoder of Alpha-CLIP to generate Alpha-CLIP spatial features. For the region-range mask of the target domain SR image, to maintain consistency with the focused region of the target domain LR image, the upsampled region-range mask is used for SR image. For the source domain HR and LR images $HR^{Source}, LR^{Source}$, due to their abundant quantity, there is no need to focus on individual instances in the image with Alpha-CLIP. Therefore, the region-range mask covering the entire image is used as input.

## 3 Experiments

### 3.1 Experiment details

The experiments validate the effectiveness of the proposed IODA method using the RealSR [2], DF2K [1], GTA [30], Cityscapes [31], and ACDC [32] datasets. Specifically, the GTA, Cityscapes, and ACDC datasets utilize downsampling methods from BasicSR [33] to generate paired LR images. For pre-trained weights, the original papers provided the real-world pre-trained weights of the SR networks (HAT [24], SRFormer [25], SAFMN [26]) on the DF2K dataset. For other datasets like GTA and Cityscapes, we conduct pre-training on these datasets using the provided source code (as explained in the table notes). For the experimental devices, a 2080ti GPU is utilized for domain adaptation training of the SFAMN and SRFormer networks. Due to the larger memory demands of the HAT network, a V100 GPU is employed as the training device. For experimental data selection, we select a single LR image from the target domain dataset for network one-shot domain adaptation training. The entire target domain dataset is used as the test set for evaluating network performance. It is worth noting that to mitigate the uncertainty in network performance caused by random sample selection, we repeat the IODA-related experiments five times. Each time, a different single sample is selected from the target domain dataset for adaptation training, and the results are presented as mean and variance.

Table 1: Effectiveness validation of IODA. Source domain dataset: DF2K [1]. Target domain dataset: RealSR_Canon [2]. Network architecture: SAFMN [26].

| Method | PSNR | SSIM |
|---|---|---|
| Baseline | 24.824 | 0.744 |
| Image-guided | 27.014 ± 0.185 | 0.773 ± 2.379e-04 |
| Image-guided & Diversity strategy (IODA) | 27.461 ± 0.008 | 0.790 ± 9.865e-06 |

### 3.2 Ablation experiment

#### 3.2.1 Effectiveness ablation experiments for each module of IODA

To validate the effectiveness of the proposed modules, ablation experiments were conducted for each module. The network pre-trained on the source domain dataset was tested on the target domain dataset, and its test results were used as the baseline.

As shown in Table 1, with the introduction of image-guided domain adaptation (Image-guided), guiding the SR network through the cross-domain direction vector between LR images in the source and target domains resulted in improvements of 2.19 in PSNR and 0.029 in SSIM performance metrics. Furthermore, with the introduction of the instance-guided target domain distribution expansion strategy (Image-guided & Diversity strategy), the diversity of the target domain distribution was further expanded, improving network performance with an increase in PSNR by 0.447 and SSIM by 0.017. Additionally, as shown in Appendix D Figure 8 and Figure 9, our proposed method effectively alleviates the negative impact of domain gaps on network performance.

Table 2: Ablation experiments on image-guided domain adaptation, using the DF2K [1] dataset as the source domain dataset and RealSR_Canon [2] as the target domain dataset, with the SAFMN network architecture [26]. The performance of the pre-trained network (pre-trained on the source domain) on the target domain is represented as the baseline. - Indicates that the corresponding domain adaptive method is ineffective for SR task.

| Method | PSNR | SSIM |
|---|---|---|
| Baseline | 24.824 | 0.744 |
| Text Guided | - | - |
| Real HR | $27.157 \pm 0.152$ | $0.787 \pm 1.988 \times 10^{-5}$ |
| Image-Guided & SR (IODA) | $27.461 \pm 0.008$ | $0.790 \pm 9.865 \times 10^{-6}$ |

### 3.2.2 Ablation experiments on image-guided domain adaptation

As shown in Table 2, we validated a text-guided domain adaptation method (Text Guided) similar to [20, 19]. In this experiments, for a clearer and more consistent representation of domain differences, we employed domain adaptation from GTA (game scenes) to Cityscapes (real scenes), with text descriptions "driving in a game" and "driving" (same as [20]). However, the collinearity constraints from text are not suitable for tasks like SR that require high-detail fidelity. They are ineffective in expressing cross-domain directions, thus unable to achieve domain adaptation for SR.

Furthermore, during domain adaptation, replacing the SR images $SR^{Source}$ generated by the source domain SR network with real HR images $HR^{Source}$ was performed to verify whether more accurate HR images would provide greater performance improvements (Real HR). However, compared to using predicted generated SR images (Image-Guided & SR), the domain adaptation network using real HR images exhibited a decrease in PSNR performance by 0.304. This indicates that using real HR images causes the target domain network to lose information from the source domain network during domain adaptation, lacking the explicit constraints of the source domain network. Besides retaining some source domain information through target domain weight initialization, no information about the source domain network is acquired during domain adaptation training. Using $SR^{Source}$ generated by source domain network can implement explicit constraints of the source domain network on the target domain network, demonstrating better domain adaptation efficiency.

Table 3: Ablation experiments on diversity strategies, using the DF2K [1] dataset as the source domain dataset and RealSR_Canon [2] as the target domain dataset, with the SAFMN network architecture [26]. The baseline refers to the domain adaptation performance after incorporating image-guided domain adaptation without any diversity enhancement.

| Method | PSNR | SSIM |
|---|---|---|
| Baseline | $27.014 \pm 0.185$ | $0.773 \pm 2.379 \times 10^{-4}$ |
| Occlusion Mask | $27.270 \pm 0.049$ | $0.790 \pm 1.865 \times 10^{-5}$ |
| Region-range Mask -G | $27.259 \pm 0.084$ | $0.788 \pm 4.396 \times 10^{-5}$ |
| Region-range Mask -B | $27.321 \pm 0.021$ | $0.791 \pm 1.450 \times 10^{-5}$ |
| Region-range Mask -I | $27.384 \pm 0.008$ | $0.788 \pm 1.897 \times 10^{-5}$ |
| Occlusion Mask + Region-range Mask -I (IODA) | $27.461 \pm 0.008$ | $0.790 \pm 9.865 \times 10^{-6}$ |

### 3.2.3 Ablation experiments on diversity strategies

As shown in Table 3, we also conducted extensive ablation experiments in the diversity strategy. Inspired by MAE [28], we directly added random occlusion masks to the images (Occlusion Mask). Furthermore, we conducted ablation experiments on the shape of region-range mask in the Alpha-CLIP network, including random grid region-range mask (Region-range Mask -G), single random-sized region-range mask (Region-range Mask -B), and instance region-range masks (Region-range Mask -I). From Table 3, it can be observed that instance region-range masks implicitly convey semantic information, leading to better domain adaptation performance compared to randomly generated region-range masks. Finally, we combined the instance region-range masks with random occlusion masking (Occlution Mask + Region-range Mask -I). From Figure 4c and Table 3, it can be observed that the proposed method effectively improved the diversity of target domain distribution and significantly enhanced the performance of the network on the target domain dataset, with a PSNR improvement of 0.447.

Table 4: Robustness experiments on different source and target domain datasets with the pre-trained network SAFMN [26]. The performance of the pre-trained network (pre-trained on the source domain) on the target domain is represented as the baseline. $\rightarrow$ signifies the domain adaptation from the source domain to the target domain.

| Method | Task | PSNR | SSIM |
|---|---|---|---|
| DF2K $\rightarrow$ RealSR_Canon | Baseline | 24.824 | 0.744 |
| | Add IODA | $27.461 \pm 0.008$ | $0.790 \pm 9.865 \times 10^{-6}$ |
| DF2K $\rightarrow$ RealSR_Nikon | Baseline | 23.980 | 0.707 |
| | Add IODA | $26.616 \pm 0.042$ | $0.757 \pm 9.852 \times 10^{-6}$ |
| GTA $\rightarrow$ Cityscapes | Baseline | 38.390 | 0.966 |
| | Add IODA | $38.938 \pm 0.002$ | $0.962 \pm 7.012 \times 10^{-8}$ |
| GTA $\rightarrow$ ACDC_rain | Baseline | 31.854 | 0.897 |
| | Add IODA | $31.919 \pm 9.891 \times 10^{-5}$ | $0.894 \pm 8.560 \times 10^{-7}$ |
| GTA $\rightarrow$ ACDC_snow | Baseline | 31.698 | 0.869 |
| | Add IODA | $31.751 \pm 2.605 \times 10^{-4}$ | $0.867 \pm 5.339 \times 10^{-7}$ |
| Cityscapes $\rightarrow$ ACDC_rain | Baseline | 28.550 | 0.879 |
| | Add IODA | $30.396 \pm 0.112$ | $0.879 \pm 1.279 \times 10^{-5}$ |
| Cityscapes $\rightarrow$ ACDC_snow | Baseline | 29.110 | 0.848 |
| | Add IODA | $30.521 \pm 0.016$ | $0.848 \pm 1.834 \times 10^{-5}$ |

### 3.2.4 Ablation experiments on different datasets

To validate the effectiveness of the proposed method across different data domains, we conducted experiments on various source and target domain datasets. From Table 4, it can be observed that the proposed method exhibits good robustness to different data distributions, showing varying degrees of performance improvement across multiple datasets.

### 3.2.5 Ablation experiments on different network architectures

To validate the effectiveness of the proposed method on different network architectures, as shown in Table 5, we conducted experiments on SAFMN [26], HAT [24], and SRFormer [25] network architectures, respectively. It was observed that IODA exhibited good robustness, showing performance improvements with one-shot domain adaptation across different network architectures.

### 3.3 Comparative experiment

In this section, we compared IODA with other methods (As shown in Table 6), including Zero-shot SR [34], domain adaptation SR network (DASR [14]), self-implemented variant of one-shot DASR, self-implemented text-guided domain adaptation methods for SR tasks (Style-GAN-NADA [19],

Table 5: Robustness experiments on different networks architectures, with the source domain dataset: DF2K and target domain dataset: RealSR_Canon. The performance of the pre-trained network (pre-trained on the source domain) on the target domain is represented as the baseline. Fine-tune refers to the network utilizing all samples from the target domain $HR^{Target}$ to fine-tune the pre-trained weights from the souce domain (Same as the original papers).

| Method | Task | PSNR | SSIM |
|---|---|---|---|
| SAFMN [26] | Baseline | 24.824 | 0.744 |
| | Fine-tune | 29.875 | 0.845 |
| | Add IODA | $27.461 \pm 0.008$ | $0.790 \pm 9.865 \times 10^{-6}$ |
| HAT [24] | Baseline | 27.477 | 0.8035 |
| | Fine-tune | 29.989 | 0.851 |
| | Add IODA | $27.821 \pm 0.008$ | $0.812 \pm 1.118 \times 10^{-6}$ |
| SRFormer [25] | Baseline | 25.946 | 0.7893 |
| | Fine-tune | 29.899 | 0.848 |
| | Add IODA | $27.657 \pm 0.045$ | $0.810 \pm 1.663 \times 10^{-5}$ |

Table 6: Performance comparison on the RealSR_Canon [2] real-world dataset. - Indicates that the corresponding domain adaptive method is ineffective for SR task. None indicates that the results are temporarily unavailable. Time represents the duration of domain adaptive training.

| Method | PSNR | SSIM | TIME(MIN) |
|---|---|---|---|
| DASR(Domain)[14] | 26.23 | 0.766 | None |
| DASR_oneshot(Domain)[14] | 23.22 | 0.645 | 19.33 |
| ZSSR[34] | 26.01 | 0.748 | 641.17 |
| Style-GAN-NADA[19] | - | - | None |
| P0DA[20] | - | - | None |
| BSRGAN[35] | 26.91 | None | None |
| Real-ESRGAN[36] | 26.14 | None | None |
| SwinIR[37] | 26.64 | None | None |
| DASR[38] | 27.40 | None | None |
| HAT[24] | 27.48 | 0.804 | None |
| SAFMN[26]+IODA | 27.46 | 0.790 | 15.3 |
| SRFormer[25]+IODA | 27.66 | 0.810 | 11.23 |
| HAT[24]+IODA | 27.82 | 0.812 | 9.01 |

P0DA [20]), classic real-world SR networks (BSRGAN [35], Real-ESRGAN [36]), and blind SR networks (Swinlr [37], DASR [38], HAT [24]). It can be observed that proposed method achieves satisfactory performance.

## 4 Conclusion

In this paper, we propose Instance-guided One-shot Domain Adaptation for SR (IODA), which enables efficient domain adaptation with only a single unlabeled target domain LR image. IODA introduces an image-guided domain adaptation method that is more effective in enforcing constraints on texture details compared to existing text-guided domain adaptation methods. Additionally, we propose the instance-guided target domain distribution expansion strategy. This strategy utilizes the diversity of instances in images to expand the diversity of domain distribution, effectively enhancing the performance of one-shot domain adaptation.

**Limitation:** Although the proposed method effectively mitigates the negative impact of the domain gap on SR network performance, domain adaptation training still requires a relatively long time (over ten minutes), which is not user-friendly for individuals who only need to super resolution of a single photo. In the future, we plan to further optimize the speed of domain adaptation, for example, by refining the selection of parameters related to network domain adaptation.

## Acknowledgments

This work was supported by the Natural Science Foundation of China (Grant 62176119), and the Jiangsu Graduate Research Innovation Program (Grant KYCX24_0259).

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

# A    Related Work

## A.1    One/Zero-shot domain adaptation

Luo et al. [21] conducted research on zero-shot domain adaptation of object detection networks from daytime to nighttime scenes. They utilized data augmentation to maximize the distance between daytime and nighttime input images while minimizing the differences in features within the backbone network, thereby enhancing the network's robustness to variations in image brightness. GAL et al. [19] were the first to integrate the CLIP network [22] into the Style-GAN [39] image generation model for zero-shot domain adaptation. They proposed a text-guided image generation network domain adaptation method, leveraging the robustness of the CLIP network in open-world scenarios to guide network adaptation from the source domain to the target domain using text descriptions. Furthermore, GAL et al.[19] proposed a strategy for selecting fine-tuning layers. During the network adaptation towards the target domain, specific layers are adjusted based on the magnitude of weight changes to enhance the stability of network domain adaptation. Kim et al. replaced the StyleGAN network in the StyleGAN-NADA method with the diffusion network [40], proposing DiffusionCLIP [41], which achieved better performance. Guo et al. [16] argued that StyleGAN-NADA and DiffusionCLIP identify all images using a single text description, leading to overly uniform directions in domain adaptation and potential pattern collapse issues. Therefore, they proposed a specific text descriptor prediction module that individually predicts a specific text description for each source domain image, effectively reducing the risk of pattern collapse. To increase the diversity of domain transfer directions, Jeon et al. [17] proposed a perturbation strategy where noise is artificially added during text feature extraction, along with an orthogonal constraint to reduce the redundancy of the noise. Additionally, Jeon et al. [17] argued that specifying certain layers for fine-tuning in StyleGAN-NADA is unreasonable, because the remaining layers also contribute to the network's predictions in the target domain. Therefore, they introduced an EWC [42] (Elastic Weight Consolidation) regularization term to fine-tune all layers of the network, but with constraints on the range of weight changes to avoid excessive adjustments. Fahes et al.[20] proposed the zero-shot domain adaptation method P0DA for semantic segmentation. Its core was a text-guided feature augmentation module. This allowed the augmentation module to augment source domain features according to the text description of the target domain, obtaining features that approximated the target domain distribution. As a result, during training, the network gained the ability to handle target domain feature distributions.

## A.2    Super resolution network

Dong et al.[43] was the first to introduce neural networks into the SR task, achieving end-to-end SR. Liang et al.[38] constructed a multi-expert system where each expert was intended to handle image information within different degraded subspace. Through cooperation among multiple experts, the system was capable of processing image information across the entire degradation space. Liang et al.[37] introduced Swin-Transformer [44] into the SR network, which effectively alleviated issues such as patch boundary artifacts and limited receptive fields with patches, using a sliding window approach. HAT [24] further enhanced the feature correlation between windows in SwinIR by introducing Overlapping Cross-Attention Blocks (OCAB) to enhance feature interactions between adjacent windows. Additionally, they combined channel attention with self-attention methods to activate more pixels for SR reconstruction, further improving its SR performance.

Although Transformer [45] could bring significant performance improvements to SR networks, the computational burden it imposed was considerable. Therefore, Zhang et al. [46] first employed shift convolution to extract local structural information from images while maintaining complexity similar to 1x1 convolutions. Then, they introduced group-wise multi-scale self-attention (GMSA) modules, which used different window sizes to compute self-attention on non-overlapping feature groups. By cascading shift convolution with GMSA modules, ELAN effectively enhanced the processing speed of existing Transformer-based SR networks. Zhou et al. [25] aimed to reduce the computational burden of networks using large-scale window self-attention. Its core concept was permuted self-attention (PSA), which reduced the channel dimension of the key and value matrices and then incorporated spatial information into the channel dimension. This approach effectively prevented the loss of spatial information while reducing the number of channels. Sun et al. [26] split features along the channel dimension and extracted these split features at different downsampling ratios, enabling the network to have a larger receptive field while effectively reducing computational cost.

## A.3 Domain Adaptation Methods in SR

The feature distribution differences between the training and test sets led to significant performance degradation of SR networks that performed well on the training set when evaluated on the test set. To address this issue, test-time domain-adaptive SR networks [34],[47],[48],[49],[50] were introduced, treating the training set as the source domain and the test set as the target domain. During inference on the target domain, the network simulated the degradation experienced by target domain LR images to generate additional training samples. Training the network with these simulated samples, which approximated the target domain's degradation, effectively reduced the negative impact of distribution discrepancies on network performance.

Shocher et al. [34] can be seen as a test-time domain-adaptive SR network. During inference, it performed Bicubic downsampling on target domain LR images to generate pseudo-LR images, simulating the degradation of the target domain LR images. It then used paired target domain LR and pseudo-LR images for supervised training, achieving super-resolution without requiring labels for target domain HR images. Soh et al. [47] suggested that the ZSSR [34] network repeatedly performed domain adaptation training from the random initial weights, leading to long training times. Therefore, they attempted to find a universal initial weight parameter to reduce the duration of domain adaptation training. Deng et al. [48] argued that the ZSSR network's consideration of only the Bicubic downsampling degradation model was insufficient to represent the more complex degradation models encountered by LR images in real-world scenarios. Therefore, they proposed the SRTTA network, which considered various degradation factors such as GaussianBlur, DefocusBlur, GlassBlur, and GaussianNoise. They used a pre-trained degradation classification network to identify the degradation category of target domain LR images and generated corresponding pseudo-LR images based on this classification. This more accurate degradation modeling enabled SRTTA to achieve better SR performance. Rad et al. [49] constrained fine-tuning samples by actively selecting additional reference samples that optimize fine-tuning efficiency, thereby improving network performance. Additionally, Zhang et al. [50] and Cheng et al. [51] applied the concept of test-time domain adaptation to propose Light Field Super-Resolution and Hyperspectral Image Super-Resolution, respectively.

Although test-time domain-adaptive SR networks had considered various degradation models, real-world scenarios involved highly complex degradation due to factors such as lighting and imaging devices, which manual degradation models could not fully represent. To address this, adversarial generated domain adaptation methods emerged [14],[52],[53],[54],[15],[55],[56],[57], using generative adversarial networks for implicit modeling of degradation, thereby avoiding the need for complex manual modeling. Fritsche et al. [53] separated high-frequency and low-frequency information for domain adaptation training. They considered that texture details correspond to high-frequency information, which is crucial for SR tasks. Therefore, they applied high-frequency filtering before feeding the features into the discriminator, using the discriminator to constrain the high-frequency information, effectively improving SR performance in reconstructing texture details. Ji et al. [54] similarly constrained generated images at the frequency level, using the discriminator for adversarial training on high-frequency information and introducing a Frequency Density Comparator to enable the network to perceive frequency differences at varying sampling rates, further improving SR performance. Subsequently, Wang et al. [15] considered the impact of domain distance between the target domain and the source domain on network domain adaptation training. They optimized the network adaptation process based on the domain distance mapped by a discriminator, assigning higher learning weights to samples with higher domain similarity, further enhancing the network's fit to the target domain. Yin et al. [55] also adopted the concept of distance awareness from [52] and achieved good performance in facial SR tasks. Xu et al. [56] introduced two adversarial adaptation modules to align source domain features with target domain features, achieving effective cross-device domain adaptive super-resolution performance.

While adversarial generated domain adaptation networks achieved good performance, they required a large number of target domain samples for network adaptation, making deployment challenging in real-world scenarios. Testing-time domain adaptation methods could perform inference on individual test samples from the target domain, but they required complex manual modeling of the target domain's degradation model and separate degradation modeling and training for each test LR image, which was time-consuming.

The Alpha-CLIP model [23], trained on millions of data samples, covers a wide range of scenarios, including various lighting conditions and degradation models, and possesses rich prior knowledge

and strong generalization. Therefore, we proposed the IODA method, which leverages Alpha-CLIP's extensive prior knowledge to guide domain adaptation for SR networks. IODA performs domain adaptation using only a single LR image from the target domain without requiring HR image labels. Additionally, when performing SR inference on a batch of data, domain adaptation training on a single LR image suffices to achieve efficient super-resolution for all LR images in the target domain.

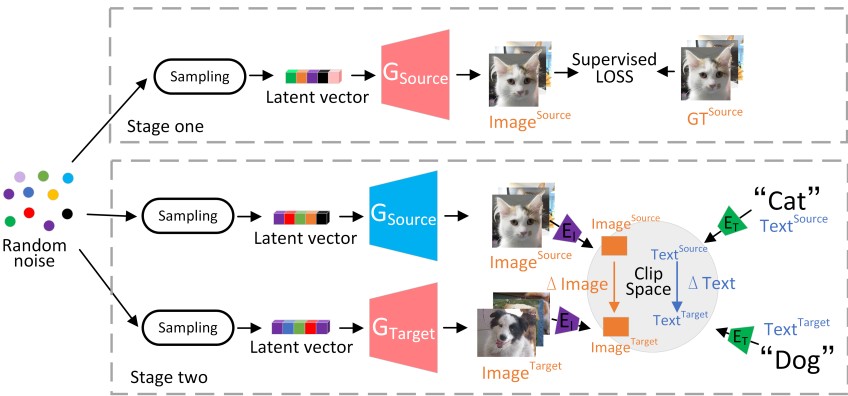

Figure 6: Text-guided domain adaptation. Existing generative networks typically sample from random noise to generate latent vectors, which are then input into the generator network to produce images. Red indicates adjustable weights, while blue indicates frozen weight.

## B   Text-guided domain adaptation

As shown in Figure 6, the existing text-guided domain adaptation is primarily used in the field of image generation. In the first stage, after pretraining the source domain generation network $G_{Source}$ using the source domain dataset, it is duplicated to serve as the target domain generation network $G_{Target}$. In the second stage, sample from random noise to generate latent vectors. Then, the source domain generation network $G_{Source}$ and the target domain generation network $G_{Target}$ are used to respectively predict the source domain generated image $Image^{Source}$ and the target domain generated image $Image^{Target}$ from latent vectors. Corresponding source domain text description $Text^{Source}$ ("Cat") and target domain text description $Text^{Target}$ ("Dog") are provided. Convert the source domain generated image, target domain generated image, source domain text description, and target domain text description into CLIP space using CLIP image encoder and text encoder, respectively. Existing methods aim to determine the text cross-domain direction vector $\Delta Text$ from source domain text features to target domain text features in CLIP space, as well as to determine the image cross-domain direction vector $\Delta Image$ from source domain generated image features to target domain generated image features in CLIP space. By aligning these two cross-domain direction vectors, training constraints are imposed on the target domain generation network. For instance, if the text changes from "cat" to "dog," the generated image should also transition from a cat image to a dog image.

## C   Alpha-CLIP

As shown in Figure 7, Alpha-CLIP [23] introduces an additional region-range masking branch (Alpha Conv) compared to CLIP [22]. During network training, this method concatenates the region-range mask with RGB images into the CLIP image encoder $E_I$ and feeds the corresponding text descriptions of the masked regions into the CLIP text encoder $E_T$. Contrastive loss is used to constrain the outputs of both encoders, where paired text-image samples are positive and unpaired samples are negative. The introduction of region-range masks allows the network to focus more on target features corresponding to the masked regions during inference.

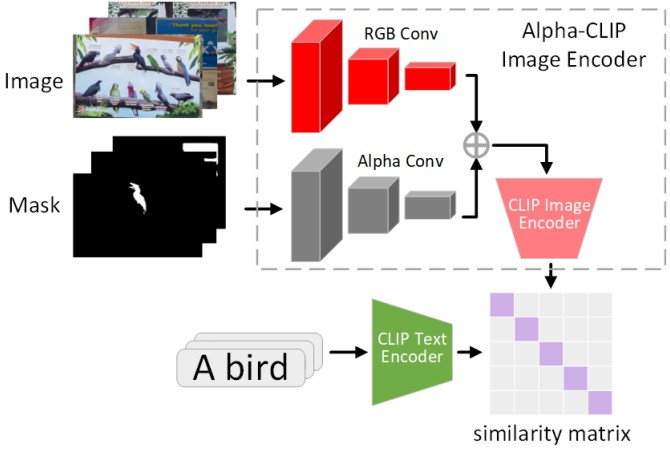

Figure 7: Alpha-CLIP

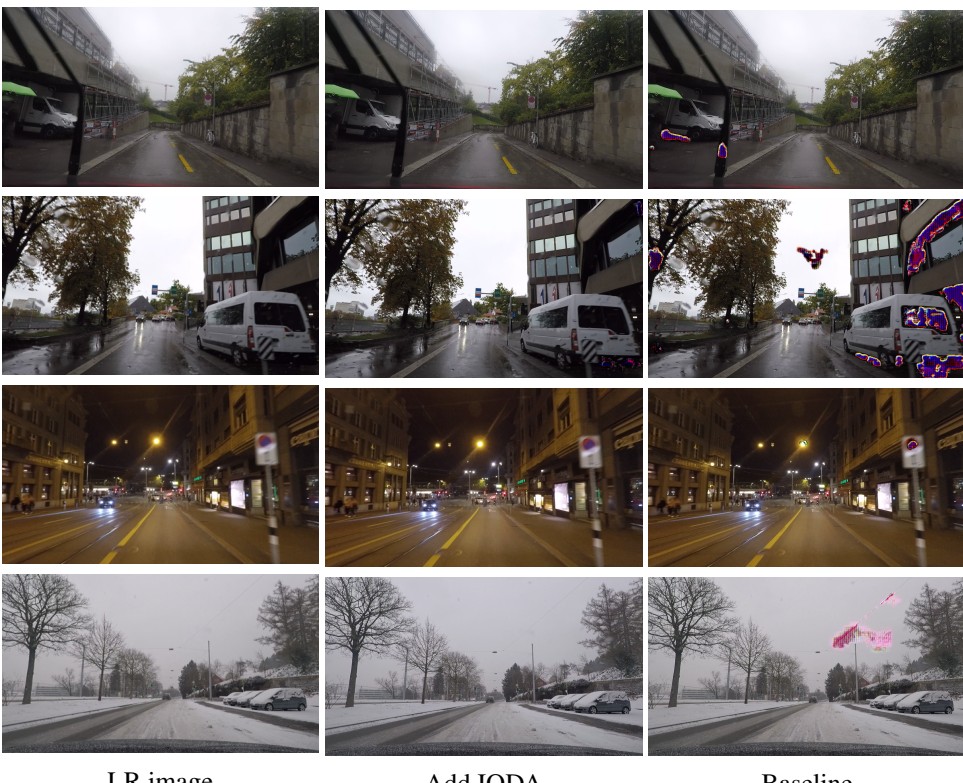

| LR image | Add IODA | Baseline |

Figure 8: Visual comparisons.The source domain dataset is the GTA [30] daytime scene dataset, and the target domain dataset includes various scene branches from the ACDC [32] dataset, such as rain, night, and snow. The network architecture used is the SAFMN [26] network.

# D    Visualization results

The network pre-trained on the source domain dataset was tested on the target domain dataset, and its test results were used as the baseline. From Figure 8, it can be observed that domain gap causes significant noises in the baseline model under rainy, night, and snowy scenes. After the SR network undergoes IODA domain adaptation, the noise is eliminated, resulting in a much improved visual appearance. In addition, as shown in Figure 9, IODA demonstrates superior performance compared to other domain adaptation methods for SR.

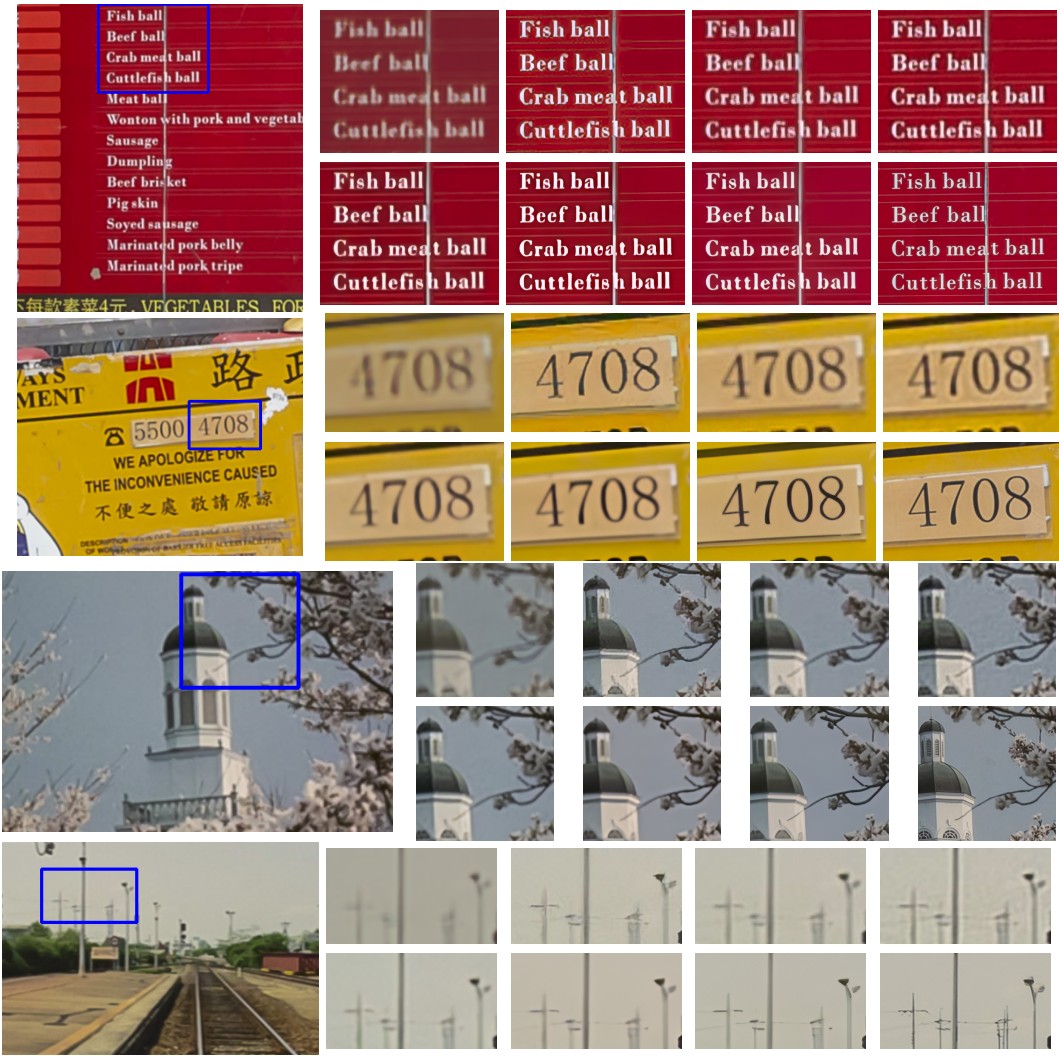

Figure 9: The large image on the left is the LR image, and the sub-images on the right are DADA [56], DASR [15], ZSSR [34], SRTTA [48] (first row), SAFMN [26] +IODA, SRFormer [25] +IODA, HAT [24] +IODA, and GT images (second row). Please zoom-in on screen.

