# OpenReview forum: "IODA: Instance-Guided One-shot Domain Adaptation for Super-Resolution"
_NeurIPS.cc/2024/Conference — NeurIPS 2024 poster_

### Official Review · Reviewer_ohtL · 2024-07-10

**Soundness:** 3
**Presentation:** 3
**Contribution:** 3
**Rating:** 5
**Confidence:** 4

**Summary:**

The author uses CLIP to assist in extracting the representation of target domain samples and implements a one-shot domain adaptation framework.

**Strengths:**

The introduction of CLIP improves the performance of one-shot domain adaptation.

**Weaknesses:**

Although the method is effective, it does not appear to be the first time that CLIP has been used for domain adaptation. Moreover, this work seems to merely combine Alpha-CLIP with SR.In addition, the training scenario for CLIP may involve cross-domain scenarios of super-resolution (SR). However, the author did not explain this issue.

**Questions:**

Why can Alpha-CLIP enhance the performance of SR when other versions of CLIP cannot?

**Limitations:**

The proposed method exhibits slow inference speed.

---

> ### Author Rebuttal · Authors · 2024-08-06
>
> **Question1:** Why can Alpha-CLIP enhance the performance of SR when other versions of CLIP cannot?
>
> **Response:** Conventional CLIP-based SR network domain adaptation methods face limited target domain diversity when using a single target domain LR image. To address this problem, we propose an instance-guided target domain distribution expansion method. Similar to human perception of the environment, which achieves comprehensive understanding by repeatedly focusing on different targets within a scene, we generate multiple image features from a single LR image, each focusing on different regions. We argue that focusing on instances within the image, rather than randomly selected regions, can implicitly introduce high-level semantic information. Thus, we optimize the focus areas to target different instances.
>
> Alpha-CLIP, a new model introduced at CVPR24, enhances the standard CLIP by allowing specified focus areas. This makes it an ideal feature extractor for our approach. We first use the SAM model to construct an instance mask pool, then select a certain number of instance masks to guide Alpha-CLIP to focus on different instances in the image, ultimately expanding the target domain feature distribution.
>
> Therefore, alpha-CLIP's unique ability to specify focus areas makes it indispensable for our proposed method.
>
>
>
> **Question2:** Although the method is effective, it does not appear to be the first time that CLIP has been used for domain adaptation. Moreover, this work seems to merely combine Alpha-CLIP with SR. In addition, the training scenario for CLIP may involve cross-domain scenarios of super-resolution (SR). However, the author did not explain this issue.
>
> **Response:** The CLIP model, trained on millions of data samples, covers a wide range of scenarios, including various lighting conditions and degradation models, and possesses rich prior knowledge and strong generalization. This makes it widely used in downstream tasks such as domain adaptation for object detection and image generation, leveraging CLIP's extensive prior knowledge and strong generalization capabilities for efficient domain adaptation. Therefore, this paper introduces CLIP for the first time in the domain adaptation for low-level tasks like super-resolution, using its rich prior information for SR domain adaptation.
>
> However, existing CLIP-based domain adaptation methods cannot be directly applied to SR tasks, as SR focuses more on restoring low-level features like texture details, while existing tasks emphasize high-level semantic information. To address this, we propose an image-guided domain adaptation method for SR tasks. Additionally, to address the limited diversity in target domain distribution caused by single target domain sample scenarios, we introduce an instance-guided target domain distribution expansion strategy. Our innovative approach efficiently enhances distribution diversity by focusing on different instances.
>
> In the first innovation, we specifically optimize domain adaptation networks for super-resolution by proposing image-guided domain adaptation instead of text-guided approaches. In the second innovation, to address the limited diversity in target domain distribution, we introduce an instance-guided diversity expansion strategy. This is the first use of Alpha-CLIP for enhancing target domain feature distribution, with a designed instance region partitioning scheme rather than random mask shapes.
>
>
>
> **Question3:** Explanation and optimization of network adaptation training time consumption
>
> **Response:** Existing domain adaptation methods for SR can be broadly classified into test-time domain adaptation methods and adversarial generated domain adaptation methods. The test-time adaptation methods need to individually manual explicit model degradation and perform domain adaptation training for each image, which can be time-consuming for relatively larger target domain datasets (see Table 2 of the Rebuttal materials) and complex to manual model.
>
> As summarized in Section 2.1 of the paper, the proposed IODA method utilizes a single LR image from the target domain for adaptation training, enabling the network to adapt from the source domain to the target domain. IODA requires only one adaptation training session on a single LR image from the target domain to perform super-resolution inference on the entire  target domain dataset, eliminating the need for repeated domain adaptation training on other target domain LR images. Additionally, IODA method leverages the rich prior semantic information from CLIP, resulting in lower training time compared to adversarial generated domain adaptation methods for SR task (DASR_oneshot and DASR (Domain)).
>
> Additionally, the domain adaptation training time of IODA can be optimized. When the source and target domains have similar distributions in high-level semantic information, the adaptation training time can be significantly reduced. For example, as shown in Table 2 of the Rebuttal materials, when the source domain is the daytime driving dataset Cityscapes and the target domain is the rainy driving dataset ACDC_rain, the training time is reduced to 1.16 minutes. In addition, selective fine-tuning of network parameters is also an effective strategy for speed improvement, which will be the focus of our future work.

---

> ### Author Response · Authors · 2024-08-13
>
> Dear Reviewer ohtL,
>
> We sincerely appreciate the time and effort you have invested in reviewing our paper.
>
> We would like to inquire whether our response has addressed your concerns and if you have the time to provide further feedback on our rebuttal. We are more than willing to engage in further discussion.
>
> Best regards,
>
> Authors of paper 4658.

---

### Official Review · Reviewer_G4LY · 2024-07-12

**Soundness:** 3
**Presentation:** 3
**Contribution:** 3
**Rating:** 5
**Confidence:** 4

**Summary:**

This paper proposes a framework of efficient instance-guided one-shot domain adaptation (abbr. IODA) with only one unlabeled target domain LR image for addressing image super-resolution (SR) issues. On top of that, it designs an instance-guided target domain distribution expansion strategy to expand the diversity of domain distribution, thus enhancing one-shot DA performance. Extensive ablation studies across multiple datasets and networks have shown the effectiveness of IODA.

**Strengths:**

1. The paper provides a new perspective for solving the problem of image super-resolution based on the domain adaptation method, especially in resource-constrained situations.

2. IODA achieves efficient domain adaptation by using a single unlabeled low-resolution (LR) image of the target domain. Moreover, the instance-guided target domain distribution expansion strategy prevents the pattern collapse by expanding the diversity of domain distribution. These facilitate IODA to carry out practical applications in the real world.

**Weaknesses:**

1. In order to clearly illustrate the methodology of this paper, the description of the specific implementation of the image-guided domain adaptation and instance-guided target domain distribution extension strategies needs to be further supplemented and refined, in particular with respect to the associated constraints.

2. The evaluation of SR results cannot be limited to pixel-oriented PSNR and SSIM, it is equally important for the evaluation of visual perceptual effects, especially in real-world applications. In addition, visual demonstrations should be provided in the main text, not just in an appendix.

**Questions:**

1. During model training, is the selection of LR images for unlabeled target domains is random? Or does it need to be based on some criteria that are not stated in this manuscript?

2. In practice, is it also necessary to add some screening criteria or conditional restrictions to the distribution expansion strategy to limit its scope? If the selection is random, how to ensure its representativeness?

**Limitations:**

While IODA provides inspiration for subsequent solutions to the task of domain adaption-based SR, it may encompass too many existing approaches such as CLIP, Alpha-CLIP, SAM, and it would further enhance the significance and value of this work if some specialized designs or optimizations could be proposed for the current research.

---

> ### Author Rebuttal · Authors · 2024-08-06
>
> **Question1:** During model training, is the selection of LR images for unlabeled target domains is random? Or does it need to be based on some criteria that are not stated in this manuscript?
>
> **Response:** To ensure the reliability of the experimental results, we repeated each experiment 5 times and took the average result, selecting different images from the dataset each time as the single LR image for training in the target domain. In the ablation studies, to avoid the impact of different target domain images on module performance, we kept the target domain images consistent across all ablation experiments, using the first 5 images from the target domain dataset.
>
>
>
> **Question2:** In practice, is it also necessary to add some screening criteria or conditional restrictions to the distribution expansion strategy to limit its scope? If the selection is random, how to ensure its representativeness?
>
> **Response:** The proposed target domain distribution expansion strategy primarily extends the CLIP spatial features by introducing Alpha-CLIP, which generates features focusing on different instances. This method does not introduce new objects into an image, unlike other data augmentation methods such as Cut-mix, which might place a cat into an image full of dogs. Instead, it ensures that the network, when generating multiple feature maps, has each feature map focus on different dogs within the image. For example, one feature map focuses on a standing dog, while the next focuses on a lying dog. This approach avoids disrupting the original feature distribution with newly introduced feature distributions.
>
> This method is similar to the human visual sensory mechanism, where one repeatedly focuses on different objects within a scene to achieve a comprehensive understanding of the scene.
>
>
>
> **Question3:** The evaluation of SR results cannot be limited to pixel-oriented PSNR and SSIM, it is equally important for the evaluation of visual perceptual effects, especially in real-world applications. In addition, visual demonstrations should be provided in the main text, not just in an appendix.
>
> **Response:** As shown in Figure 1 of the Rebuttal materials, we provide additional visualizations to demonstrate the effectiveness of the proposed method.
>
>
>
> **Question4:** Regarding the use of CLIP, Alpha-CLIP, and SAM.
>
> **Response:** In this paper, we address the specificity challenges of domain adaptation for low-level tasks like super-resolution by proposing an image-guided domain adaptation method. This method leverages the focus on detailed textures in super-resolution tasks, using images to guide the adaptation process and overcoming the limitations of text-guided approaches in representing fine details. Additionally, to address the limited diversity in target domain distribution caused by single target domain sample scenarios, we introduce an instance-guided target domain distribution expansion strategy. Our innovative approach efficiently enhances distribution diversity by focusing on different instances.
>
> In the first innovation, we specifically optimize domain adaptation networks for super-resolution by proposing image-guided domain adaptation instead of text-guided approaches. In the second innovation, to address the limited diversity in target domain distribution, we introduce an instance-guided diversity expansion strategy. This is the first use of Alpha-CLIP for enhancing target domain feature distribution, with a designed instance region partitioning scheme rather than random mask shapes.
>
> Many excellent works utilize large models like CLIP, Llama, and SAM as the foundation for further exploration. For example, current domain adaptation networks for object detection use the CLIP network for guidance, but we do not deny the contributions of their work. Similarly, this paper employs powerful models like Alpha-CLIP and SAM as tools to achieve our objectives.

---

> ### Author Response · Authors · 2024-08-13
>
> Dear Reviewer G4LY,
>
> We sincerely appreciate the time and effort you have invested in reviewing our paper.
>
> We would like to inquire whether our response has addressed your concerns and if you have the time to provide further feedback on our rebuttal. We are more than willing to engage in further discussion.
>
> Best regards,
>
> Authors of paper 4658.

---

### Official Review · Reviewer_V6qe · 2024-07-12

**Soundness:** 3
**Presentation:** 3
**Contribution:** 3
**Rating:** 5
**Confidence:** 4

**Summary:**

The paper presents a novel approach to one-shot domain adaptation for super resolution. The key idea is to use the CLIP directional vector between low resoultion source and target domain images to guide the SR image generation in the target domain. They further use occlusion masks to further increase the performance of the model by providing pre-trained Alpha-CLIP with different region-range masks, enabling it to generate Alpha-CLIP spatial features focused on different areas of the image, thereby expanding the diversity of the target domain distribution.

**Strengths:**

- The paper has a well-organized structure and expresses its ideas clearly.
- Ablation analysis is provided in the paper to study the effect of various components of the pipeline on the downstream performance.
- The idea is pretty interesting and new and the motivation is sound.

**Weaknesses:**

- Given the high time complexity of the approach, the performance gains over the baseline seems to be small.
- The SR performance of the model needs to be compared against stronger SR baselines to better analyze the performance gains.
- More visualization analysis is needed in the paper to better support the performance gains.

**Questions:**

- Can you compare the performance of your models against the other domain adaptation-based SR approaches like [12] and [13]?
- Can you provide a more comprehensive visualization analysis to better support you performance claims?
- Can you provide a comparison of the efficiency between this method and other methods? A comparison that considers both efficiency and PSNR/SSIM would be more reasonable.





[12] Yunxuan Wei, Shuhang Gu, Yawei Li, Radu Timofte, Longcun Jin, and Hengjie Song. Unsuper304 vised real-world image super resolution via domain-distance aware training. In 2021 IEEE/CVF Conference on Computer Vision and Pattern Recognition (CVPR), pages 13380–13389, 2021.
13] Wei Wang, Haochen Zhang, Zehuan Yuan, and Changhu Wang. Unsupervised real-world super307 resolution: A domain adaptation perspective. In 2021 IEEE/CVF International Conference on Computer Vision (ICCV), pages 4298–4307, 2021.

**Limitations:**

The main limitations of this paper is the relatively low training efficiency that would impose certain limitations in practical scenarios. Although the idea looks nice, but as mentioned in the paper, it takes approx 10 mins to generate a SR version of a single image. Therefore, I'm not completely convinced about its complexity.

---

> ### Author Rebuttal · Authors · 2024-08-06
>
> **Question1:** Can you compare the performance of your models against the other domain adaptation-based SR approaches like [12] and [13]?
>
> **Response:** As shown in Table 1 of the Rebuttal materials  (Rebuttal.pdf), we have additionally included domain adaptive methods for super-resolution, DADA [1] and SRTTA [2], in our comparative experiments. Currently, we compared super-resolution domain adaptive methods DASR [3], DADA [1], SRTTA [2], and ZSSR [4]. Notably, we compared the performance of DASR with single target domain LR image and multiple LR images. Additionally, we also evaluated the performance of general domain adaptive methods on SR tasks (Style-GAN-NAN [5], P0DA [6]). Reference [13] did not release its source code, so it was not included in the comparison experiments.
>
> **Question2:** Can you provide a more comprehensive visualization analysis to better support you performance claims?
>
> **Response:** As shown in Figure 1 of the Rebuttal materials  (Rebuttal.pdf), we provide additional visualizations to demonstrate the effectiveness of the proposed method.
>
> **Question3:** Can you provide a comparison of the efficiency between this method and other methods? A comparison that considers both efficiency and PSNR/SSIM would be more reasonable.
>
> **Response:** As shown in Table 1 of the Rebuttal materials  (Rebuttal.pdf), we have additionally included inference efficiency metrics to demonstrate the effectiveness of the method.
>
> **Question4:** Explanation and optimization of network adaptation training time consumption
>
> **Response:** Existing domain adaptation methods for SR can be broadly classified into test-time domain adaptation methods and adversarial generated domain adaptation methods. The test-time adaptation methods need to individually manual explicit model degradation and perform domain adaptation training for each image, which can be time-consuming for relatively larger target domain datasets (see Table 2 of the Rebuttal materials) and complex to manual model.
>
> As summarized in Section 2.1 of the paper, the proposed IODA method utilizes a single LR image from the target domain for adaptation training, enabling the network to adapt from the source domain to the target domain. IODA requires only one adaptation training session on a single LR image from the target domain to perform super-resolution inference on the entire  target domain dataset, eliminating the need for repeated domain adaptation training on other target domain LR images. Additionally, IODA method leverages the rich prior semantic information from CLIP, resulting in lower training time compared to adversarial generated domain adaptation methods for SR task (DASR_oneshot and DASR (Domain)).
>
> Additionally, the domain adaptation training time of IODA can be optimized. When the source and target domains have similar distributions in high-level semantic information, the adaptation training time can be significantly reduced. For example, as shown in Table 2 of the Rebuttal materials, when the source domain is the daytime driving dataset Cityscapes and the target domain is the rainy driving dataset ACDC_rain, the training time is reduced to 1.16 minutes. In addition, selective fine-tuning of network parameters is also an effective strategy for speed improvement, which will be the focus of our future work.

---

> ### Author Response · Authors · 2024-08-13
>
> Dear Reviewer V6qe,
>
> We sincerely appreciate the time and effort you have invested in reviewing our paper.
>
> We would like to inquire whether our response has addressed your concerns and if you have the time to provide further feedback on our rebuttal. We are more than willing to engage in further discussion.
>
> Best regards,
>
> Authors of paper 4658.

---

### Official Review · Reviewer_wDY8 · 2024-07-13

**Soundness:** 3
**Presentation:** 2
**Contribution:** 3
**Rating:** 3
**Confidence:** 4

**Summary:**

This paper addresses one-shot domain adaptation (OSDA) in the field of super-resolution (SR). It leverages the fact that the content remains unchanged during super-resolution to propose an image-guided domain adaptation method, ensuring consistency by aligning the direction between the source domain and the target domain. The authors highlight the difficulty in learning the target domain's distribution during OSDA in SR and propose increasing data samples using random masking, similar to MAE. They also utilize SAM and Alpha Clip to obtain instance-aware representations. The method trains to align the direction between the source and target domains using the representation obtained from Alpha Clip. Various experiments demonstrate the effectiveness of the proposed method.

**Strengths:**

- They propose an OSDA method for real-world applications of super-resolution (SR).
- They effectively utilize foundation models to address the domain adaptation (DA) challenges in the SR task.

**Weaknesses:**

There seems to be an insufficient survey of domain adaptation methods in the SR task, specifically those outlined in references [1-4]. Major revision is needed to emphasize the necessity and originality of the proposed method, based on an analysis of these existing methods. Particularly, since [4] addresses test-time adaptation, it is essential to highlight the advantages of one-shot domain adaptation over test-time adaptation. Additionally, the experimental tables predominantly report performance improvements over the baseline, but a comparison with existing domain adaptation methods is also required.

[1] Wang, Wei, et al. "Unsupervised real-world super-resolution: A domain adaptation perspective." Proceedings of the IEEE/CVF International Conference on Computer Vision. 2021.

[2] Wei, Yunxuan, et al. "Unsupervised real-world image super resolution via domain-distance aware training." Proceedings of the IEEE/CVF conference on computer vision and pattern recognition. 2021.

[3] Xu, Xiaoqian, et al. "Dual adversarial adaptation for cross-device real-world image super-resolution." Proceedings of the IEEE/CVF Conference on Computer Vision and Pattern Recognition. 2022.

[4] Deng, Zeshuai, et al. "Efficient test-time adaptation for super-resolution with second-order degradation and reconstruction." Advances in Neural Information Processing Systems 36 (2023): 74671-74701.

**Questions:**

Please refer to the Weaknesses section

**Limitations:**

They've addressed the limitations of the proposed method.

---

> ### Author Rebuttal · Authors · 2024-08-06
>
> **Question1**: There seems to be an insufficient survey of domain adaptation methods in the SR task, specifically those outlined in references [1-4]. Major revision is needed to emphasize the necessity and originality of the proposed method, based on an analysis of these existing methods. Particularly, since [4] addresses test-time adaptation, it is essential to highlight the advantages of one-shot domain adaptation over test-time adaptation.
>
> **Response:** In the paper, we describe domain adaptation for SR tasks in Appendix A1, "One/Zero-shot Domain Adaptation", where references 1 and 2 cited by the Reviewer correspond to references 12 and 13 in the paper. To provide a clearer and more comprehensive overview of domain adaptation methods in the SR task, we will add a section 'Domain Adaptation Methods in SR' in the related work. In this section, we added descriptions of references [3, 4], provided a detailed analysis of test-time domain adaptation networks, and highlighted the advantages of one-shot domain adaptation.
>
> **Domain Adaptation Methods in SR**
>
> The feature distribution differences between the training and test sets led to significant performance degradation of SR networks that performed well on the training set when evaluated on the test set. To address this issue, test-time domain-adaptive SR networks were introduced, treating the training set as the source domain and the test set as the target domain. During inference on the target domain, the network simulated the degradation experienced by target domain LR images to generate additional training samples. Training the network with these simulated samples, which approximated the target domain's degradation, effectively reduced the negative impact of distribution discrepancies on network performance.
>
> Shocher et al. [28] can be seen as a test-time domain-adaptive SR network. During inference, it performed Bicubic downsampling on target domain LR images to generate pseudo-LR images, simulating the degradation of the target domain LR images. It then used paired target domain LR and pseudo-LR images for supervised training, achieving super-resolution without requiring labels for target domain HR images. Deng et al. [4] argued that the ZSSR network's consideration of only the Bicubic downsampling degradation model was insufficient to represent the more complex degradation models encountered by LR images in real-world scenarios. Therefore, they proposed the SRTTA network, which considered various degradation factors such as GaussianBlur, DefocusBlur, GlassBlur, and GaussianNoise. They used a pre-trained degradation classification network to identify the degradation category of target domain LR images and generated corresponding pseudo-LR images based on this classification. This more accurate degradation modeling enabled SRTTA to achieve better SR performance.
>
> Although test-time domain-adaptive SR networks had considered various degradation models, real-world scenarios involved highly complex degradation due to factors such as lighting and imaging devices, which manual degradation models could not fully represent. To address this issue, adversarial generated domain adaptation methods emerged, using generative adversarial networks for implicit modeling of degradation, thus avoiding complex manual modeling.
>
> Wang et al. [13] employed a generative network to generate fake LR images paired with high-resolution (HR) images and used a discriminator to constrain the generated LR images to align with the target domain distribution. Subsequently, Wei et al. [12] considered the impact of domain distance between the target domain and the source domain on network domain adaptation training. They optimized the network adaptation process based on the domain distance mapped by a discriminator, assigning higher learning weights to samples with higher domain similarity, further enhancing the network’s fit to the target domain. Xu et al. [3] introduced two adversarial adaptation modules to align source domain features with target domain features, achieving effective cross-device domain adaptive super-resolution performance.
>
> While adversarial generated domain adaptation networks achieved good performance, they required a large number of target domain samples for network adaptation, making deployment challenging in real-world scenarios. Testing-time domain adaptation methods could perform inference on individual test samples from the target domain, but they required complex manual modeling of the target domain’s degradation model and separate degradation modeling and training for each test image, which was time-consuming.
>
> The CLIP model, trained on millions of data samples, covers a wide range of scenarios, including various lighting conditions and degradation models, and possesses rich prior knowledge and strong generalization. Therefore, we proposed the IODA method, which leverages CLIP’s extensive prior knowledge to guide domain adaptation for SR networks. IODA performs domain adaptation using only a single LR image from the target domain without requiring HR image labels. Furthermore, when performing SR inference on a batch of data, domain adaptation training is required only for the first image, enabling efficient inference for subsequent images.
>
> **Question2**: Additionally, the experimental tables predominantly report performance improvements over the baseline, but a comparison with existing domain adaptation methods is also required
>
> **Response:**  As shown in Table 1 of the Rebuttal materials, we have additionally included domain adaptive methods for super-resolution, DADA and SRTTA, in our comparative experiments. Currently, we compared super-resolution domain adaptive methods DASR, DADA, SRTTA, and ZSSR. Notably, we compared the performance of DASR with single target domain samples and multiple data samples. Additionally, we also evaluated the performance of general domain adaptive methods (StyleGAN-NAN, P0DA).

---

> ### Author Response · Authors · 2024-08-13
>
> Dear Reviewer wDY8,
>
> We sincerely appreciate the time and effort you have invested in reviewing our paper.
>
> We would like to inquire whether our response has addressed your concerns and if you have the time to provide further feedback on our rebuttal. We are more than willing to engage in further discussion.
>
> Best regards,
>
> Authors of paper 4658.

---

> > ### Comment · Reviewer_wDY8 · 2024-08-13
> >
> > Thank you for the responses, but I still believe that this paper requires a major revision, and my concerns are as follows:
> >
> > In the introduction section, it is stated that existing methods need to train on many low-resolution (LR) images from the target domain (L39) and that there is no work addressing this issue (L53). However, the authors should carefully survey and compare related studies, including those I mentioned in my initial review, to demonstrate the necessity and distinctiveness of this research. The current survey does not seem thorough enough, and the advantages of this work compared to the mentioned studies are not very convincing. The paper should be reorganized with a focus on addressing these issues.

---

> > > ### Author Response · Authors · 2024-08-14
> > > **Revised survey on SR domain adaptation [1/3]**
> > >
> > > Dear Reviewer,
> > >
> > > 1. We have further optimized the section on "domain adaptation for SR."
> > >
> > > 2. It is worth noting that domain adaptation for SR primarily addresses the issue of unpaired SR, with the main approach based on adversarial networks for domain alignment.  Due to the nature of adversarial networks, they require a substantial number of samples for training. While test-time domain adaptation networks have optimizations for training time, they still require repeated domain adaptation training for each image in the target domain, leading to relatively longer processing times. Our proposed IODA utilizes the rich prior knowledge of Alpha-CLIP for domain adaptation guidance, **requiring adaptation training on only a single LR image from the target domain and eliminating the need for repeated degradation modeling and training on all images.**
> > >
> > >
> > > ## **Domain Adaptation Methods in SR**
> > >
> > > The feature distribution differences between the training and test sets led to significant performance degradation of SR networks that performed well on the training set when evaluated on the test set. To address this issue, test-time domain-adaptive SR networks [1,2,3,4,5] were introduced, treating the training set as the source domain and the test set as the target domain. During inference on the target domain, the network simulated the degradation experienced by target domain LR images to generate additional training samples. Training the network with these simulated samples, which approximated the target domain's degradation, effectively reduced the negative impact of distribution discrepancies on network performance.
> > >
> > > Shocher et al. [1] can be seen as a test-time domain-adaptive SR network. During inference, it performed Bicubic downsampling on target domain LR images to generate pseudo-LR images, simulating the degradation of the target domain LR images. It then used paired target domain LR and pseudo-LR images for supervised training, achieving super-resolution without requiring labels for target domain HR images. Soh et al. [2] suggested that the ZSSR [1] network repeatedly performed domain adaptation training from the random initial weights, leading to long training times.  Therefore, they attempted to find a universal initial weight parameter to reduce the duration of domain adaptation training. Deng et al. [3] argued that the ZSSR network's consideration of only the Bicubic downsampling degradation model was insufficient to represent the more complex degradation models encountered by LR images in real-world scenarios. Therefore, they proposed the SRTTA network, which considered various degradation factors such as GaussianBlur, DefocusBlur, GlassBlur, and GaussianNoise. They used a pre-trained degradation classification network to identify the degradation category of target domain LR images and generated corresponding pseudo-LR images based on this classification. This more accurate degradation modeling enabled SRTTA to achieve better SR performance. Rad et al. [4] constrained fine-tuning samples by actively selecting additional reference samples that optimize fine-tuning efficiency, thereby improving network performance. Additionally, Zhang et al. [5] and Cheng et al. [6] applied the concept of test-time domain adaptation to propose Light Field Super-Resolution and Hyperspectral Image Super-Resolution, respectively.

---

> > > ### Author Response · Authors · 2024-08-14
> > > **Revised survey on SR domain adaptation [2/3]**
> > >
> > > Although test-time domain-adaptive SR networks had considered various degradation models, real-world scenarios involved highly complex degradation due to factors such as lighting and imaging devices, which manual degradation models could not fully represent. To address this, adversarial generated domain adaptation methods emerged [7,8,9,10,11,12,13,14,15,16], using generative adversarial networks for implicit modeling of degradation, thereby avoiding the need for complex manual modeling. Wang et al. [7], Sun et al. [8] and Cong  et al. [9] employed a generative network to generate fake LR images paired with high-resolution (HR) images and used a discriminator to constrain the generated LR images to align with the target domain distribution. Subsequently, Fritsche et al. [10] separated high-frequency and low-frequency information for domain adaptation training.  They considered that texture details correspond to high-frequency information, which is crucial for SR tasks. Therefore, they applied high-frequency filtering before feeding the features into the discriminator, using the discriminator to constrain the high-frequency information, effectively improving SR performance in reconstructing texture details. Ji et al. [11] similarly constrained generated images at the frequency level, using the discriminator for adversarial training on high-frequency information and introducing a Frequency Density Comparator to enable the network to perceive frequency differences at varying sampling rates, further improving SR performance. Huang et al.[12] proposed a RGB image guided infrared super-resolution network, effectively reducing the negative impact of RGB image noise on infrared super-resolution performance through frequency-domain constraints. Subsequently, Wang et al. [13] considered the impact of domain distance between the target domain and the source domain on network domain adaptation training. They optimized the network adaptation process based on the domain distance mapped by a discriminator, assigning higher learning weights to samples with higher domain similarity, further enhancing the network’s fit to the target domain. Yin et al. [14] also adopted the concept of distance awareness from [8] and achieved good performance in facial SR tasks. Xu et al. [15] introduced two adversarial adaptation modules to align source domain features with target domain features, achieving effective cross-device domain adaptive super-resolution performance.
> > >
> > > While adversarial generated domain adaptation networks achieved good performance, they required a large number of target domain samples for network adaptation, making deployment challenging in real-world scenarios. Testing-time domain adaptation methods could perform inference on individual test samples from the target domain, but they required complex manual modeling of the target domain’s degradation model and separate degradation modeling and training for each test LR image, which was time-consuming.
> > >
> > > The Alpha-CLIP model [17], trained on millions of data samples, covers a wide range of scenarios, including various lighting conditions and degradation models, and possesses rich prior knowledge and strong generalization. Therefore, we proposed the IODA method, which leverages  Alpha-CLIP’s extensive prior knowledge to guide domain adaptation for SR networks. IODA performs domain adaptation using only a single LR image from the target domain without requiring HR image labels. Additionally, when performing SR inference on a batch of data, domain adaptation training on a single LR image suffices to achieve efficient super-resolution for all LR images in the target domain.

---

> > > ### Author Response · Authors · 2024-08-14
> > > **Revised survey on SR domain adaptation [3/3]**
> > >
> > > [1] A. Shocher, N. Cohen and M. Irani, "Zero-Shot Super-Resolution Using Deep Internal Learning," 2018 IEEE/CVF Conference on Computer Vision and Pattern Recognition, Salt Lake City, UT, USA, 2018, pp. 3118-3126, doi: 10.1109/CVPR.2018.00329.
> > >
> > > [2] J. W. Soh, S. Cho and N. I. Cho, "Meta-Transfer Learning for Zero-Shot Super-Resolution," *2020 IEEE/CVF Conference on Computer Vision and Pattern Recognition (CVPR)*, Seattle, WA, USA, 2020, pp. 3513-3522, doi: 10.1109/CVPR42600.2020.00357.
> > >
> > > [3] Deng, Zeshuai, et al. "Efficient test-time adaptation for super-resolution with second-order degradation and reconstruction." Advances in Neural Information Processing Systems 36 (2023): 74671-74701.
> > >
> > > [4] M. S. Rad, T. Yu, B. Bozorgtabar and J. -P. Thiran, "Test-Time Adaptation for Super-Resolution: You Only Need to Overfit on a Few More Images," *2021 IEEE/CVF International Conference on Computer Vision Workshops (ICCVW)*, Montreal, BC, Canada, 2021, pp. 1845-1854, doi: 10.1109/ICCVW54120.2021.00211.
> > >
> > > [5] L. Zhang, J. Nie, W. Wei and Y. Zhang, "Unsupervised Test-Time Adaptation Learning for Effective Hyperspectral Image Super-Resolution With Unknown Degeneration," in *IEEE Transactions on Pattern Analysis and Machine Intelligence*, vol. 46, no. 7, pp. 5008-5025, July 2024, doi: 10.1109/TPAMI.2024.3361894.
> > >
> > > [6] Z. Cheng, Z. Xiong, C. Chen, D. Liu and Z. -J. Zha, "Light Field Super-Resolution with Zero-Shot Learning," *2021 IEEE/CVF Conference on Computer Vision and Pattern Recognition (CVPR)*, Nashville, TN, USA, 2021, pp. 10005-10014, doi: 10.1109/CVPR46437.2021.00988.
> > >
> > > [7] Y. Wei, S. Gu, Y. Li, R. Timofte, L. Jin and H. Song, "Unsupervised Real-world Image Super Resolution via Domain-distance Aware Training," *2021 IEEE/CVF Conference on Computer Vision and Pattern Recognition (CVPR)*, Nashville, TN, USA, 2021, pp. 13380-13389, doi: 10.1109/CVPR46437.2021.01318.
> > >
> > > [8] W. Sun, D. Gong, Q. Shi, A. van den Hengel and Y. Zhang, "Learning to Zoom-In via Learning to Zoom-Out: Real-World Super-Resolution by Generating and Adapting Degradation," in *IEEE Transactions on Image Processing*, vol. 30, pp. 2947-2962, 2021, doi: 10.1109/TIP.2021.3049951.
> > >
> > > [9] S. Cong , K. Cui , Y.Yang , Y. Zhou,  X. Wang, H Luo,  Y Zhang, X Yao, "DDASR: Domain-Distance Adapted Super-Resolution Reconstruction of MR Brain Images," [J]. medRxiv, 2023: 2023.06. 29.23292026.
> > >
> > > [10] M. Fritsche, S. Gu and R. Timofte, "Frequency Separation for Real-World Super-Resolution," *2019 IEEE/CVF International Conference on Computer Vision Workshop (ICCVW)*, Seoul, Korea (South), 2019, pp. 3599-3608, doi: 10.1109/ICCVW.2019.00445.
> > >
> > > [11] X. Ji, “Frequency Consistent Adaptation for Real World Super Resolution”, *AAAI*, vol. 35, no. 2, pp. 1664-1672, May 2021.
> > >
> > > [12] Huang Y, Miyazaki T, Liu X, et al. Target-oriented domain adaptation for infrared image super-resolution[J]. arXiv preprint arXiv:2311.08816, 2023.
> > >
> > > [13] Wang, Wei, et al. "Unsupervised real-world super-resolution: A domain adaptation perspective." Proceedings of the IEEE/CVF International Conference on Computer Vision. 2021.
> > >
> > > [14] Z. Yin, M. Liu, X. Li, H. Yang, L. Xiao and W. Zuo, "MetaF2N: Blind Image Super-Resolution by Learning Efficient Model Adaptation from Faces," *2023 IEEE/CVF International Conference on Computer Vision (ICCV)*, Paris, France, 2023, pp. 12987-12998, doi: 10.1109/ICCV51070.2023.01198.
> > >
> > > [15] Xu, Xiaoqian, et al. "Dual adversarial adaptation for cross-device real-world image super-resolution." Proceedings of the IEEE/CVF Conference on Computer Vision and Pattern Recognition. 2022.
> > >
> > > [16] P. Albert *et al*., "Unsupervised domain adaptation and super resolution on drone images for autonomous dry herbage biomass estimation," *2022 IEEE/CVF Conference on Computer Vision and Pattern Recognition Workshops (CVPRW)*, New Orleans, LA, USA, 2022, pp. 1635-1645, doi: 10.1109/CVPRW56347.2022.00170.
> > >
> > > [17] Sun Z, Fang Y, Wu T, et al. Alpha-clip: A clip model focusing on wherever you want[C]//Proceedings of the IEEE/CVF Conference on Computer Vision and Pattern Recognition. 2024: 13019-13029.

---

> ### Author Response · Authors · 2024-08-13
>
> Thank you for your response.
>
> 1. I have included the four papers you mentioned in the newly added related work section. Our original related work, which covers SR and zero/one-shot domain adaptation, is in the appendix of the original paper. With our previous reply, we have now included related work for all three branches.
>
> 2. Regarding domain adaptation in SR, the reference [2] you mentioned is the first to introduce domain adaptation into super-resolution. Most domain adaptation work in SR uses a Cycle-GAN-like architecture, which, due to its adversarial nature, requires a large number of target domain samples for training. As a result, there has been little work focusing on single-target domain samples and incorporating CLIP-based models to guide SR domain adaptation.
>
> 3. Compared to existing multi-sample-based SR domain adaptation networks, the proposed method demonstrates better adaptability and performance for single target domain samples (Rebuttal.pdf, Table1: DASR_oneshot, DASR_Domain,DADA). Unlike test-time domain adaptation networks, our IODA method leverages CLIP's rich prior knowledge to perform domain adaptation training on a single target domain sample and efficiently infer SR for all remaining samples, without requiring repeated domain adaptation training for each LR image in the target domain (Rebuttal.pdf, Table1,2: SRTTA,ZSSR). Additionally, in the newly added experiments (Rebuttal.pdf), we validate this point, showing that IODA performs well in both domain adaptation training time and performance metrics.

---

### Author Rebuttal · Authors · 2024-08-06

1. ### **Additional visual demonstrations**

As shown in Figure 1 of the Rebuttal materials (Rebuttal.pdf), we provide additional visualizations to demonstrate the effectiveness of the proposed method.

2. ### **Additional comparisons with domain adaptive methods for Super-resolution**

As shown in Table 1 of the Rebuttal materials  (Rebuttal.pdf), we have additionally included domain adaptive methods for super-resolution, DADA [1] and SRTTA [2], in our comparative experiments. Currently, we compared super-resolution domain adaptive methods DASR [3], DADA [1], SRTTA [2], and ZSSR [4]. Notably, we compared the performance of DASR with single target domain LR image and multiple LR images. Additionally, we also evaluated the performance of general domain adaptive methods on SR tasks (Style-GAN-NAN [5], P0DA [6]).

3. ### **Explanation and optimization of network adaptation training time consumption**

Existing domain adaptation methods for SR can be broadly classified into test-time domain adaptation methods and adversarial generated domain adaptation methods. The test-time adaptation methods need to individually manual explicit model degradation and perform domain adaptation training for each image, which can be time-consuming for relatively larger target domain datasets (see Table 2 of the Rebuttal materials) and complex to manual model.

As summarized in Section 2.1 of the paper, the proposed IODA method utilizes a single LR image from the target domain for adaptation training, enabling the network to adapt from the source domain to the target domain. IODA requires only one adaptation training session on a single LR image from the target domain to perform super-resolution inference on the entire  target domain dataset, eliminating the need for repeated domain adaptation training on other target domain LR images. Additionally, IODA method leverages the rich prior semantic information from CLIP, resulting in lower training time compared to adversarial generated domain adaptation methods for SR task (DASR_oneshot and DASR (Domain)).

Additionally, the domain adaptation training time of IODA can be optimized. When the source and target domains have similar distributions in high-level semantic information, the adaptation training time can be significantly reduced. For example, as shown in Table 2 of the Rebuttal materials, when the source domain is the daytime driving dataset Cityscapes and the target domain is the rainy driving dataset ACDC_rain, the training time is reduced to 1.16 minutes. In addition, selective fine-tuning of network parameters is also an effective strategy for speed improvement, which will be the focus of our future work.



###  **Reference**

[1] Xu, Xiaoqian, et al. "Dual adversarial adaptation for cross-device real-world image super-resolution." Proceedings of the IEEE/CVF Conference on Computer Vision and Pattern Recognition. 2022.

[2] Deng, Zeshuai, et al. "Efficient test-time adaptation for super-resolution with second-order degradation and reconstruction." Advances in Neural Information Processing Systems 36 (2023): 74671-74701.

[3] Wang, Wei, et al. "Unsupervised real-world super-resolution: A domain adaptation perspective." Proceedings of the IEEE/CVF International Conference on Computer Vision. 2021.

[4] A. Shocher, N. Cohen and M. Irani, "Zero-Shot Super-Resolution Using Deep Internal Learning," 2018 IEEE/CVF Conference on Computer Vision and Pattern Recognition, Salt Lake City, UT, USA, 2018, pp. 3118-3126, doi: 10.1109/CVPR.2018.00329.

[5] Gal R, Patashnik O, Maron H, et al. Stylegan-nada: Clip-guided domain adaptation of image generators[J]. ACM Transactions on Graphics (TOG), 2022, 41(4): 1-13.

[6] M. Fahes, T. -H. Vu, A. Bursuc, P. Pérez and R. De Charette, "PØDA: Prompt-driven Zero-shot Domain Adaptation," *2023 IEEE/CVF International Conference on Computer Vision (ICCV)*, Paris, France, 2023, pp. 18577-18587, doi: 10.1109/ICCV51070.2023.01707.

---

### Decision · Program_Chairs · 2024-09-25

**Decision:**

Accept (poster)

**Comment:**

This paper presents an approach to using the CLIP directional vector to guide the SR image generation in the target domain. The method is interesting and recognized by most reviewers. The authors have addressed most reviewer concerns, receiving three borderline accept recomendations. The remaining concerns mainly from wDY8, who believes the authors should provide a more convincing survey of existing methods. In the rebuttal, the authors have updated the survey, which make clear the difference between the new method and domain adaption methods in SR.